# Characterization of Clinical MRSA Isolates from Northern Spain and Assessment of Their Susceptibility to Phage-Derived Antimicrobials

**DOI:** 10.3390/antibiotics9080447

**Published:** 2020-07-25

**Authors:** Marina Salas, Maciej Wernecki, Lucía Fernández, Beatriz Iglesias, Diana Gutiérrez, Andrea Álvarez, Laura García, Elisabeth Prieto, Pilar García, Ana Rodríguez

**Affiliations:** 1DairySafe Group, Instituto de Productos Lácteos de Asturias (IPLA-CSIC), Paseo Río Linares s/n, Villaviciosa, 33300 Asturias, Spain; marinasalasf@gmail.com (M.S.); lucia.fernandez@ipla.csic.es (L.F.); Diana.GutierrezFernandez@UGent.be (D.G.); anarguez@ipla.csic.es (A.R.); 2Institute of Genetics and Microbiology, Faculty of Biological Sciences, University of Wroclaw, 50-120 Wroclaw, Poland; maciej.wernecki@uwr.edu.pl; 3Instituto de Investigación Sanitaria del Principado de Asturias (ISPA), Oviedo, 33011 Asturias, Spain; 4Servicio de Microbiología, Hospital San Agustín, Avilés, 33401 Asturias, Spain; beatriz.iglesias@sespa.es (B.I.); laura.garcia@sespa.es (L.G.); melisabeth.prieto@sespa.es (E.P.); 5Laboratory of Applied Biotechnology, Department of Applied Biosciences, Faculty of Bioscience Engineering, Ghent University, 9000 Ghent, Belgium; 6Servicio de Medicina Interna, Hospital San Agustín, Avilés, 33401 Asturias, Spain; andrea.alvarez@sespa.es

**Keywords:** MRSA, bacteriophages, endolysins, hospital infections, biofilms, virulence genes

## Abstract

Methicillin-resistant *Staphylococcus aureus* (MRSA) is a prevalent nosocomial pathogen, causing a wide range of diseases. The increased frequency of MRSA isolates in hospitals and the emergence of vancomycin resistance have sparked the search for new control strategies. This study aimed to characterize sixty-seven MRSA isolates collected from both infected patients and asymptomatic carriers in a Spanish hospital. RAPD-PCR allowed the identification of six genetic patterns. We also investigated the presence of genes involved in producing adhesins, toxins and the capsule; the biofilm; and antimicrobial resistance. A notable percentage of the isolates carried virulence genes and showed medium-high ability to form biofilms. Next, we assessed the strains’ susceptibility to two phages (phiIPLA-C1C and phiIPLA-RODI) and one endolysin (LysRODI). All strains were resistant to phiIPLA-C1C, and most (70.2%) were susceptible to phiIPLA-RODI. Regarding LysRODI, all strains displayed susceptibility, although to varying degrees. There was a correlation between endolysin susceptibility and the random amplification of polymorphic DNA (RAPD) profile or the presence of some virulence genes (*fnbA*, *eta*, *etb*, *PVL* and *czr*), but that was not observed with biofilm-forming ability, strain origin or phage sensitivity. Taken together, these findings can help to explain the factors influencing endolysin effectiveness, which will contribute to the development of efficient therapies targeting MRSA infections.

## 1. Introduction

The pathogenicity of *Staphylococcus aureus* relies on its capacity to synthetize an important number of toxins and virulence factors, together with its ability to form biofilms [1]. On top of that, there is an upward trend in the selection and spread of antibiotic resistance determinants amongst *S. aureus* strains. All these characteristics allow this bacterium to successfully evade host defenses and conventional antibiotic treatment strategies [2]. It is also worth highlighting the adaptability of this pathogen, which might, to some extent, be due to its genome plasticity. Indeed, approximately 25% of the *S. aureus* chromosome consists of mobile genetic elements (MGEs), such as pathogenicity islands, bacteriophages, chromosomal cassettes, transposons and plasmids, all of which can be acquired by horizontal transfer between strains. Consequently, gain and loss of virulence and antimicrobial resistance determinants may contribute to bacterial adaptability, virulence and survival in the face of environmental challenges [3]. Perhaps in recent years, the most concerning aspect of *S. aureus’s* epidemiology has been the extensive distribution of methicillin-resistant *S. aureus* (MRSA) strains, which represent a serious threat to healthcare worldwide. Indeed, MRSA strains are now endemic in many American and European hospitals (hospital-associated MRSA strains, HA-MRSA), accounting for 29–35% of all clinical isolates [4]. Moreover, the fact that *S. aureus* can easily adapt to multiple environments has likely contributed to the appearance of community-associated (CA-MRSA) and livestock-associated (LA-MRSA) strains, which have evolved independently from HA-MRSA strains. In terms of global epidemiology, HA-MRSA is endemic in hospitals; CA-MRSA is spreading within the general population; and LA-MRSA is present in farms [5]. One interesting feature of MRSA epidemiology is that, despite the general diversity of this pathogen, relatively few clonal lineages are dominant. Moreover, an evolution with time is frequently observed, so that, in a specific geographical area, the predominant lineages are periodically replaced by new epidemic clones [6]. Additionally, the distribution and incidence of the prevalent clones differ between geographical regions [5]. In terms of clinical pathology, HA-MRSA strains cause pneumonia, bacteremia, endocarditis and bone infections. Bacteremia and endocarditis are the most serious infections, with mortality rates that can reach up to 60% [7]. HA-MRSA is a very common cause of ventilator-associated pneumonia (VAP) and hospital-associated pneumonia (HAP), probably the most frequent infections in critically ill patients, with a mortality of 3–17% [8]. *S. aureus* is also the most common etiological agent of acute hematogenous osteomyelitis (AHO) in pediatric patients [9]. Moreover, MRSA, MSSA (methicillin-sensitive *S. aureus*) and coagulase-negative staphylococci are the most frequent pathogens in surgical site infections after total hip and knee arthroplasty [10].

Today, the successful treatment of MRSA infections remains an unsolved challenge that requires the evaluation of novel antimicrobials [11]. In this context, the application of bacteriophages and phage-derived lytic proteins represents a promising strategy for both the treatment and prophylaxis of MRSA infections. Indeed, bacteriophages (or phages) are viruses that exclusively infect bacteria, and can, therefore, be considered as their natural killers. Although phages have been used to fight infectious diseases in Eastern Europe since the early 20th century (phage therapy), it is only in the last decade that they have resurged as an alternative against multidrug-resistant bacteria [12]. Moreover, phage lytic proteins (endolysins) are also promising candidates as new antimicrobials. Endolysins are phage-encoded peptidoglycan hydrolases that act during the latter stages of the lytic cycle to break down the cell wall, and as a result, lyse the host cell to release the newly formed viral particles. These proteins represent a novel class of antibiotics named enzybiotics that selectively and rapidly kill specific Gram-positive bacteria when added externally to cells [13]. In addition, the potential for development of bacterial resistance is considered very low, due probably to the fact that they target the highly conserved peptidoglycan bonds [14]. For several endolysins, a synergistic effect with antibiotics has also been demonstrated [15]. Additionally, the efficacy of endolysins has been studied in various animal models of infection with promising results [16]. Regarding their safety and toxicity, a detailed study carried out with two pneumococcal endolysins, Pal and Cpl-1, reported no change in gene expression for human macrophages and pharyngeal cells exposed to endolysins. Likewise, mice injected with these proteins exhibited no physical or behavioral changes, no hypersensitivity or allergic reaction, constant pro-inflammatory cytokine levels and no significant changes in the fecal microbiome [17]. Moreover, the main drawback of endolysins, i.e., their lack of activity against Gram-negative bacteria, has already been overcome by developing fusion proteins that consist of an endolysin plus a cationic peptide (Artilysin^®^) [18] or an endolysin plus a bacteriocin (lysocins) [19]. Given their enormous therapeutic potential, there is a growing number of studies regarding the activity of endolysins against *S. aureus* strains. For example, endolysin SAL-1, derived from the *Staphylococcus*-specific bacteriophage SAP-1 (the active pharmaceutical ingredient of SAL200) is being evaluated in clinical trials for the treatment of antibiotic-resistant staphylococcal infections [20]. Additionally, endolysin CF-301 (ContraFect) is now in phase 2 clinical trials for bacteremia and endocarditis treatment [21].

In our previous work, we identified and characterized four *S. aureus*-infecting phages: two siphophages, phiIPLA88 and phiIPLA35, and two myophages, phiIPLA-RODI and phiIPLA-C1C [22,23]. Characterization of phiIPLA-RODI revealed that it is a broad-host-range phage, able to infect several species belonging to the *Staphylococcus* genus [23]. Moreover, genomic analysis of this phage led to the identification of an endolysin, LysRODI, which contains two catalytic domains (CHAP and amidase-2 domain) and one cell wall-binding domain (CBD), SH3b [23,24]. Our laboratory has already demonstrated the potential of several phage endolysins to remove *S. aureus* cells from different environments. For example, various endolysins and chimeric proteins showed lytic activity against *S. aureus* cells in pasteurized milk [25,26] and biofilms [24,27,28]. Moreover, we showed that subinhibitory doses of some endolysins can inhibit biofilm formation in *S. aureus* through the downregulation of autolysin-encoding genes [28]. More recently, we showed the ability of some of these proteins (LysRODI, LysA72 and CHAPSH3b) to remove staphylococcal strains *in vivo*. Indeed, LysRODI and CHAPSH3b were non-toxic in a zebrafish embryo model and significantly reduced mortality in a zebrafish model of systemic infection [24]. Moreover, LysRODI demonstrated great efficacy in preventing mammary infections by *S. aureus* and *S. epidermidis* in a mouse model of mastitis [24]. However, most of these studies were carried out with MSSA strains. In the present work, we characterized a set of MRSA strains, some of which were isolated from patients with active infections and some from asymptomatic carriers. Then, we assessed their susceptibility to bacteriophages phiIPLA-RODI and phiIPLA-C1C, and endolysin LysRODI. We further aimed to determine whether there was any correlation between different traits of these strains and their susceptibility to this protein. Ultimately, we think that this information will help us to reach our long-term goal of optimizing the use of phage-derived endolysins to treat MRSA infections.

## 2. Materials and Methods

### 2.1. Bacterial Strains and Culture Conditions 

All the MRSA strains (Table 1) were isolated by the Microbiology Service at San Agustín Hospital (Avilés, Asturias, Spain) from November 2018 to October 2019. Isolation was carried out on blood agar medium and methicillin resistance was determined by the vitek2 system (Biomerieux). Strains were routinely grown in tryptic soy broth (TSB; Difco, Franklin Lakes, NJ) at 37 °C with shaking at 200 rpm or on plates containing TSB supplemented with 2% (wt/vol) bacteriological agar (TSA).

### 2.2. DNA Preparation and Genomic Analysis 

Total DNA from the MRSA strains was extracted by using the GenElute Bacterial Genomic DNA Kit (Sigma-Aldrich, Madrid, Spain) according to the manufacturer’s recommendations and while supplementing the lysis solution with 5 mg/mL lysozyme and 0.2 mg/mL lysostaphin. The DNA concentration and the A_260_/A_280_ purity ratio were determined by using a Nanophotometer™ (Implen GmbH, Munich, Germany).

The detection of virulence and antimicrobial resistance genes was carried out by PCR in a thermocycler (Bio-Rad, Hercules, CA) using Taq 2 × Master Mix RED, 1.5 mM MgCl_2_ (Ampliqon A/S, Odense, Denmark) and the primers shown in Table 2. All PCR products were resolved by electrophoresis in 2% agarose gels, stained with EZ-Vision^®^ One (VWR, Barcelona, Spain) and visualized under UV light with a G:BOX gel documentation system (Syngene, Cambridge, UK) equipped with GeneSys image acquisition software (Syngene). A 1-kb DNA ladder (Nippon Genetics, Dueren, Germany) was included in all gels.

Random amplification of polymorphic DNA (RAPD) analysis of the different isolates was performed using primer OPL5 (5’-ACGCAGGCAC-3’) according to a previously described method [29]. The resulting RAPD-PCR band patterns were then analyzed by agglomerative hierarchical clustering using the unweighted pair group method with arithmetic averages (UPGMA) based on the Jaccard similarity coefficient. The cophenetic correlation coefficient was calculated to determine the goodness of fit of the dendrogram. The selection of cut-off points for defining clusters was based on the default values determined by SciPy 1.0.0, and the shapes of the dendrograms.

### 2.3. Biofilm Formation Assays 

Biofilms were grown in 96-well microtiter plates (Thermo Scientific, NUNC, Madrid, Spain) according to the method described by Herrera et al. [30] with some modifications. Briefly, overnight cultures of *S. aureus* were diluted in TSBg (TSB supplemented with 0.25% *w*/*v*
d-(+)-glucose) to obtain cell suspensions each containing 10^6^ CFU/mL; 0.2 mL aliquots of this suspension were used to inoculate each well; 0.2 mL of TSBg was added to the control well. These microtiter plates were then incubated for 24 h at 37 °C. Following incubation, the planktonic phase was removed; the adhered phase was washed twice with phosphate-buffered saline (PBS) (137 mM NaCl, 2.7 mM KCl, 10 mM Na_2_HPO_4_ and 2 mM KH_2_PO_4_; pH 7.4) and subsequently stained with crystal violet. Staining was performed by adding 0.2 mL of 0.1% (*w*/*v*) crystal violet to each well. Following 15 min of incubation at room temperature, the excess dye was removed by washing twice with water. The crystal violet attached to the well was distained with 0.2 mL of 33% (*v*/*v*) acetic acid and absorbance at 595 nm was quantified with a Bio-Rad Benchmark plus microplate spectrophotometer (Bio-Rad Laboratories, Hercules, CA, USA). This experiment was performed with three independent biological replicates.

### 2.4. Quantification of Specific Lytic Activity 

Protein LysRODI was expressed and purified as described previously [23]. Turbidity reduction assays were performed using MRSA cells suspended in NaPi buffer (50 mM; pH = 7.4) and treated with two-fold dilutions of the purified protein (0.02–50 µM). Those data allowed calculation of the specific lytic activity (ΔOD_600_ × min^−1^ × µM^−1^). The results were then divided by the specific activity of the reference strain *S. aureus* Sa9 to calculate the relative specific lytic activity. This allowed the correction of differences in the specific activity values that might have been due to the use of different protein stocks. All experiments were performed in triplicate. 

### 2.5. Phage Susceptibility 

The ability of phages phiIPLA-RODI and phiIPLA-C1C to infect and lyse the different isolates was analyzed by performing spot tests. Briefly, 1:10 dilutions from overnight cultures of the different strains were used to inoculate a lawn in semisolid agar on top of a TSA plate. Once the top layer containing the bacterial cells was set, 10 μL droplets of each phage suspension were placed onto the plate and allowed to air dry. Then, the plates were incubated overnight at 37 °C and the results were visualized the following day. A strain was considered susceptible (S) only if a clear halo was visible; otherwise, the strain was considered resistant (R).

### 2.6. Data Analysis 

Data analysis and representation were performed using modules NumPy 1.15.0, pandas 0.23.3, SciPy 1.0.0, matplotlib (v.2.0.2) and scikit-learn 0.19.2 in Python 3.5.3. The independence of two categorical variables was assessed using the Chi-square test. Clustering of the biofilm and the relative specific lytic activity data was carried out with the k-means algorithm.

## 3. Results

### 3.1. Isolation of MRSA Strains at San Agustín Hospital

Throughout the period under study, 67 MRSA isolates were isolated from specimens taken from patients with ages between 17 and 95 years old, with the mean age being 75 years. The MRSA-positive individuals included 37 (44.78%) males and 30 (55.22%) females. Isolates were collected from 40 outpatients (59.70%) and 27 inpatients (40.30%). Among those MRSA isolates, 38.81% (25 strains) came from ulcer exudates, 14.93% (eight strains) from wound exudates, 11.94% (seven strains) from nasal exudates, 10.45% (seven strains) from urine samples, 5.97% (four strains) from eschar exudates, 5.97% (four strains) from abscesses, 4.48% (three strains) from blood cultures, 2.99% (two strains) from catheters, 1.49% (one strain) from conjunctival exudate, 1.49% (one strain) from pharyngeal exudate and 1.49% (one strain) from sputum (Table 1). It must be noted that 43 strains came from people with staphylococcal infections, while 22 were isolated from asymptomatic carriers (Table 1). In relation to the type of MRSA strain, 56 corresponded to hospital-associated strains, while nine were community acquired (Table 1). Regarding antibiotic resistance, most isolates (61) were resistant to levofloxacin, 11 were resistant to clindamycin and only one was resistant to trimetropim/sulfametoxazol (Table 1).

Analysis of the independence between these characteristics revealed an association between the existence or not of an infection and the sample type (Chi-square = 103.236, degrees of freedom (dof) = 20, *p* value = 3.304 × 10^−13^). For instance, all nasal, eschar and pharyngeal specimens were taken from asymptomatic carriers, whereas all blood and abscess samples came from infected patients (Figure 1A). There was also a link between the origin of the isolates and the type of clinical sample (Chi-square = 72.076, degrees of freedom (dof) = 20, *p* value = 8.315 × 10^−8^). In this case, the proportion of strains from certain types of specimens also differed between community-associated and hospital-associated isolates (Figure 1B). No association was observed between the existence or not of an active infection and the origin of the isolates, nor was there any link between antibiotic resistance and any of these features.

### 3.2. Genetic Diversity of Human MRSA Isolates 

To investigate the relationship between isolates, we first performed genomic fingerprinting analysis by RAPD-PCR. Amplification with primer OPL5 revealed the presence of six different band patterns, with bands ranging between 200 bp and 1700 bp in size (Figure 2A). The relative abundances of the six RAPD profiles (A, B, C, D, E and F) varied between the analyzed samples, with A being the most frequent (62.69% of the samples). The relative frequencies of profiles B, C, D, E and F were 16.42%, 13.43%, 2.99%, 2.99% and 1.49%, respectively. The band patterns of the different RAPD profiles were used to perform hierarchical clustering, whose results were subsequently plotted as a dendrogram. Considering a cut-off value of 0.399, the profiles appeared clustered in one large cluster that could be further subdivided in two groups, with one including profiles A, F and D and the other corresponding to profiles E and B (Figure 1A). Profile C was an orphan cluster (Figure 2A). The cophenetic correlation coefficient was 0.787, indicating that the dendrogram was a good representation of the dissimilarities between the observed band patterns. 

To further characterize the MRSA isolates, we examined the presence of genes encoding different virulence factors and antimicrobial resistance determinants (Table 1). The entire collection of isolates was phenotypically resistant to methicillin and harbored the *mec*A gene. By contrast, none of them carried the *qacA/B* gene, and only four strains (5.97%) possessed gene *czr*. We also identified the presence of genes encoding microbial surface components recognizing adhesive matrix molecules (MSCRAMMs). All 67 isolates were positive for genes *eno*, *clfA* and *clfB*. In turn, genes *bbp*, *cna*, *ebpS*, *fnbA*, *fnbB* and *fib* were respectively present in 1.49%, 16.42%, 74.63%, 13.43%, 11.94% and 82.09% of all strains. Regarding genes involved in capsule formation, all strains analyzed, except MRSA-IPLA40, carried the *cap5* gene (98.51%), while only three (MRSA-IPLA 40, 42 and 43; 4.48%) possessed gene *cap8*. Additionally, some genes involved in biofilm formation (*icaA* and *icaD*) were present in 100% of the strains analyzed, while none had the *bap* gene. Regarding toxin-encoding genes, only *tst* was present in 100% of the strains, whereas *sea* and *seb* were not detected in any isolate. The rest of the toxin-encoding genes, namely, *sec*, *sed*, *see*, *seg*, *sei*, *eta*, *etb* and PVL, were respectively carried by 5.97%, 19.40%, 8.96%, 85.07%, 85.07%, 73.13%, 62.69% and 62.69% of all strains.

Using the data regarding the RAPD profile and the presence or absence of the genes that showed variation between strains, we attempted new clustering of the isolates. In this case, the strains were grouped into three clusters when using a cut-off of 0.58 (Figure 2B). The green (cluster I) and red (cluster II) clusters consisted mostly of the nine strains with the C RAPD profile, together with the two strains with profile D (one in the green cluster and one in the red cluster) and one with profile E in the green cluster. In turn, the cyan cluster (cluster III) included all the strains with RAPD profiles A, B and F, and one strain with profile E. Therefore, cluster III corresponds largely to the large cluster of Figure 2A, with the exception of a few strains from the low frequency RAPD profiles, which are included in clusters I and II. The cophenetic correlation coefficient was 0.725, indicating that the dendrogram was a good representation of the genetic differences observed between the strains.

In order to infer whether there was any relationship between the origins of the strains and their genetic profiles, we examined the independence between their RAPD profile and other variables (type of clinical sample, active infection, origin (community/hospital), antibiotic resistance). For most variables, this analysis showed that there was no significant correlation with the RAPD profile, the only exception being levofloxacin resistance (Chi-square = 23.568, degrees of freedom (dof) = 5, *p* value = 0.0003). Thus, all isolates with RAPD profiles D and F were resistant to this antibiotic, whereas isolates with the E profile were susceptible (Figure 3).

### 3.3. Biofilm Formation Ability 

Following genetic characterization, we analyzed the abilities of the MRSA clinical strains to form biofilms on a polystyrene surface. The biomass values obtained for the 67 strains ranged between 0.555 (MRSA-IPLA 36) and 2.583 (MRSA-IPLA 56), so it was considered that all strains were capable of forming biofilms, although with different strengths. The strains were then grouped into three clusters, namely, weak (biomass ≤ 1.143), average (biomass between 1.143 and 1.861) and strong (biomass ≥ 1.861) biofilm formers, by using the k-means algorithm (Figure 4A). Strains with poor abilities to develop biofilms (cluster 0) represented 44.78% of the total sample. The second largest cluster was number 1 (41.79%), which consisted of isolates with average biofilm forming capacity. Finally, cluster 2, including the strong biofilm-forming strains, corresponded to just 13.43% of all isolates.

Next, we studied the independence between biofilm-forming strength (weak, average or strong) and previously characterized parameters (type of sample, infection, origin, RAPD profile, presence of different genes and antibiotic resistance) by performing a Chi-square test. Most of these variables showed no significant co-dependence with biofilm development. The only exception was the presence of gene *fnbA* (*p* value = 0.011). To get a closer look at this potential correlation, we examined the frequency of *fnbA* in the three biofilm clusters (Figure 4B). The results of this analysis indicated that this gene was only present in weak (26.67%) and strong (11.11%) biofilm formers but not in average biofilm-forming isolates.

### 3.4. Susceptibility to Phages phiIPLA-RODI and phiIPLA-C1C 

The results obtained in the phage susceptibility experiments revealed that all strains were resistant to phage phiIPLA-C1C, while the impact of phiIPLA-RODI varied between isolates (Table 1). Thus, 29.85% of the strains were resistant to the phage and 70.15% were susceptible. The existence of potential co-dependence between phiIPLA-RODI susceptibility and other parameters (type of sample, infection, origin, RAPD profile, biofilm formation, presence of different genes and antibiotic resistance) was examined, but no potential correlation was found.

### 3.5. Susceptibility to Endolysin LysRODI 

The susceptibility of the different MRSA isolates to endolysin LysRODI was determined by carrying out the turbidity reduction assay, which generally gives more precise results than the MIC assay for this type of antimicrobials. To do that, we added a concentration of 13.44 μg/mL (0.25 μM) of LysRODI per well, which led to a clear turbidity reduction in the reference strain *S. aureus* Sa9. The results of these assays allowed for the calculation of the specific lytic activity for each strain; those values were then divided by that of strain Sa9 to determine the relative specific lytic activity. The values obtained are shown in Table 1, and they demonstrate that all isolates exhibited lysis by LysRODI, but with different degrees of susceptibility. These values were then used to classify the strains into three clusters corresponding to isolates with low (cluster 0), average (cluster 1) and high (cluster 2) susceptibility (Figure 5A). The largest clusters, corresponding to 62.69% and 31.34% of all isolates, included strains with low and average endolysin susceptibility, respectively. By contrast, only 5.97% of the isolates were more susceptible to LysRODI than the reference strain Sa9. 

As we did with the biofilm data, we examined whether there was independence between endolysin susceptibility and other parameters (type of sample, infection, origin, RAPD profile, biofilm formation, presence of the different genes analyzed and antibiotic resistance). Again, most of the variables analyzed and endolysin susceptibility were independent according to the result of the Chi-square test. Nonetheless, others were not independent from LysRODI susceptibility, including the RAPD profile (*p* value = 0.022), *fnbA* (*p* value = 0.007), *eta* (*p* value = 0.002), *etb* (*p* value = 0.024), PVL (*p* value = 0.024) and *czr* (*p* value = 0.0002). For this reason, we assessed more in detail the frequency distribution of these parameters in the three groups of susceptibility. In the case of the RAPD profiles, strains with low susceptibility exhibited RAPD A (69.05%), B (19.05%) or C (11.90%) patterns, whereas strains with high susceptibility had RAPD A (50%), RAPD D (25%) or RAPD E (25%) profiles (Figure 5B). The cluster corresponding to average susceptibility included isolates with all six RAPD profiles. It is worth noting that low frequency profiles (F, D and E) exhibited average or high susceptibility, while more frequent RAPD profiles (A, B and C) largely had low or average susceptibility. The percentage of strains carrying *fnbA* for each cluster was 7.14%, 14.28% and 75% for strains with low, average and high susceptibility, respectively (Figure 5C). The relative frequency distribution of strains with and without genes *eta*, *etb* and *PVL* was very similar (Figure 5D–F). For example, in the case of *eta*, the gene was present in strains exhibiting low or average endolysin susceptibility at 80.95% and 71.43%, respectively. Conversely, none of those three genes was detected in the four strains with high susceptibility. Finally, gene *czr* was absent from all strains with low LysRODI susceptibility, whereas some strains with average (9.52%) and high (50%) susceptibility did carry the gene (Figure 5G).

## 4. Discussion

Infections caused by multidrug-resistant MRSA strains are becoming more and more frequent in hospitals all over the world. This represents a serious threat to healthcare practice, especially considering the immunological vulnerability of many hospitalized patients and their difficulty with fighting infections that are practically untreatable with conventional antimicrobials. Therefore, there is an urgent need for new therapeutic agents and control strategies to prevent the transmission and development of these infections. Amongst the multiple strategies aimed at combatting drug-resistant bacteria, the use of phage-based therapies represents a feasible, safe alternative to more conventional drugs. Within this context, the present work sought to evaluate the antimicrobial activities of two staphylococcal phages (phiIPLA-RODI and phiIPLA-C1C) and one endolysin (LysRODI) against MRSA strains from clinical origin, with a view to their future therapeutic use.

The first part of this study involved the collection of 67 MRSA isolates over a 12-month period in a medium-size hospital (428 beds) from different types of clinical specimens. In spite of the samples’ diverse clinical origins, the isolates seemed to be genetically related based on their RAPD-PCR patterns and virulence gene content, with one large cluster including most of the strains, and two smaller clusters comprising strains with RAPD profile C and a few isolates with low frequency profiles. The predominant genetic pattern was RAPD profile A, although no correlation could be found between a specific genetic pattern and the origin of the sample from which it was isolated. This lack of genetic diversity is not surprising, as about 90% of all studied *S. aureus* MRSA genomes can be grouped into four major clonal complexes (CC5, CC8, CC398 and CC30) [37]. Overall, all the strains had a good arsenal of virulence genes, with some of them bearing important implications in disease prognosis, such as the different toxins. Moreover, most of these isolates exhibited not only resistance to methicillin but also resistance to other antibiotics, especially levofloxacin. The accumulation of resistance determinants in this pathogen is very frequent, and is making treatment of staphylococcal infections an increasingly difficult task.

Additionally, we studied the abilities of these strains to form biofilms. This characteristic, widely spread amongst *S. aureus* strains, may negatively contribute to bacteria eradication, allowing the pathogen to survive in different environments and cause chronic infections. Somewhat unsurprisingly, all the analyzed strains were able to form biofilms on polystyrene surfaces. Infections caused by bacteria living in biofilms are more difficult to treat because cells embedded within a matrix are more resistant to antibiotics and the immune system than planktonic cells. The biofilm matrix prevents antibiotics from reaching their target due to low diffusion. Moreover, the altered metabolism of sessile bacteria makes them more tolerant to antibiotics [38]. Given that all the isolates carried the *ica* genes but not the *bap* gene, we can hypothesize that the biofilm matrix of these strains might be mainly composed of polysaccharides [39]. Additionally, most of the isolates carried other virulence factors involved in the binding of bacteria to biotic surfaces, like human tissues. Due to their role in biofilm formation on host tissues, possession of these proteins further increases the difficulty of infection treatment. As a result, they are considered potential drug targets to curtail the development of chronic infections [40]. Although still concerning from a clinical standpoint, it is worth noting that strong biofilm-forming strains represented the lowest percentage amongst the strains examined in this study. Interestingly, two strong biofilm formers, 38 and 49, were isolated from catheters, in which their good biofilm-forming capability may help them to successfully colonize medical implants. 

Bacteriophages have already been proven successful for the treatment of MRSA infections affecting skin and soft tissues [41], septicaemia [42] and wounds [43]. Here, we tested the lytic activities of two myophages, phiIPLA-RODI and phiIPLA-C1C, against the 67 MRSA clinical isolates. *Myoviridae* phages generally exhibit a broad host range and are, therefore, preferred candidates for phage therapy [44]. None of the strains were susceptible to phage phiIPLA-C1C, which is not surprising as this phage predominantly infects *S. epidermidis* strains [23]. In contrast, 70% of all isolates displayed susceptibility to phiIPLA-RODI. In a previous study, this phage showed activity against 21 *S. aureus* strains as well several strains belonging to other staphylococci [23]. Although this phage can be potentially used in phage cocktails or in combination with antibiotics to kill susceptible strains, it still would not work by itself against all MRSA isolates. 

Compared to bacteriophages, endolysins have the advantage of exhibiting a wider spectrum of activity [12]. To date, several works have demonstrated the high *in vitro* and *in vivo* activity of endolysins against *S. aureus* isolates from different origins, including clinical staphylococci (revised by Gutiérrez et al. [45]). The lytic protein used in this study, LysRODI, had been previously shown to be highly active against different staphylococcal strains [24]. Along the same lines, the data presented here indicates that all clinical MRSA isolates were susceptible to LysRODI, although to different degrees. Similarly, the recombinant endolysin HY-133 was highly active against all tested MSSA and MRSA isolates, including isolates resistant to mupirocin, ceftaroline/ceftobiprole and borderline oxacillin [46,47]. To further examine the impact of endolysin treatment, the strains were clustered into three susceptibility groups (low, average and high), the high susceptibility group being the least numerous. In contrast, while still being lysed by the enzyme, most strains belonged to the low susceptibility cluster. The degree of endolysin susceptibility in the analyzed strains had a correlation with the RAPD profile, and with the presence of genes *fnbA*, *eta*, *etb*, *PVL* and *czr*. However, this correlation is not necessarily indicative of causality; that is, the presence or absence of these genes does not necessarily determine the ability of LysRODI to lyse cells of a given strain. Nonetheless, the percentage of strains carrying *eta*, *etb* and *PVL* was considerably higher in the low susceptibility cluster. In contrast, *czr* was only present in strains with average or high susceptibility. It would be interesting to examine whether these trends are specific to this isolate collection. If not, the possible involvement of these genes in endolysin susceptibility should be studied in detail. So far, the factors that determine the different degree of endolysin susceptibility remain largely unknown. However, changes in the cell wall composition or structure are known to influence endolysin activity [47]. For instance, it has been shown that WTA composition has an impact on the efficacy of endolysin PRF-119. In fact, this is the reason why this endolysin is active against *S. aureus* strains producing polyribitol phosphate (RboP) wall teichoic acids (WTA), but not against CoNS strains, which produce polyglycerol phosphate (GroP) WTA [48]. Moreover, Idelevich et al. [49] found that *S. aureus* strains producing GroP WTA were less susceptible to PRF-119 than other strains. Modification of the cell wall structure has also been shown to influence susceptibility to other antimicrobials. For example, deletion of the *msaABCR* operon, which reduces cell wall thickness, resulted in decreased resistance to vancomycin in vancomycin-intermediate *S. aureus* (VISA). Simultaneously, this mutation also led to reduced cross-linking due to increased murein hydrolase activity and nonspecific processing of murein hydrolases. This defect was enhanced by a decrease in teichoic acid content in the *msaABCR* mutant [50].

The broad range of action of LysRODI is very promising for the treatment of staphylococcal infections, especially those due to antimicrobial resistant strains. As mentioned above, most of the isolates collected in this study were resistant to levofloxacin, clindamycin and/or trimethoprim/sulfamethoxazole in addition to methicillin. In our collection, only four strains carried the cadmium and zinc resistance gene *czr*. However, one of them was isolated from a wound. It is known that the presence of this gene hinders the efficacy of zinc-based topical agents for the treatment of staphylococcal infections [51]. Nonetheless, this strain showed an average susceptibility to endolysin LysRODI, thereby making this protein a good candidate to be used as topical treatment for wound infections. Indeed, topical application of another endolysin, S25-3, has been shown to decrease the number of intraepidermal staphylococci and the sizes of pustules in an experimental mouse model of impetigo [52]. Additionally, *S. aureus* is one of the prevalent causes of pneumonia, a serious and often difficult to treat disease because of antibiotic resistance. Here, we isolated two MRSA strains from pharyngeal and sputum samples, both of which were sensitive to both phiIPLA-RODI and LysRODI, and eight strains from nasal exudates were susceptible to the endolysin. In a previous work, Xia et al. [53] found that the activity of endolysin LysGH15 could be increased by combining it with apigenin, a natural flavonoid from fruits and vegetables, for the treatment of MRSA-caused pneumonia in mice. The two main characteristics of phage endolysins are their high specificity towards their target bacteria and the lack of bacterial resistance selection after repeated exposure. Both properties make them suitable decolonizing compounds, alternatives to mupirocin, with minimum effects on the surrounding microbiota. Indeed, decolonization of patients colonized with MRSA proved to be very useful for lowering their risk of infection [54]. Endolysins also exhibit properties that can be very useful for the treatment of chronic infections. For instance, there is *in vitro* evidence of the efficacy of some endolysins against small colony variants (SCVs) of clinical origin, such as endolysin HY-133 [55], or against persister cells, such as LysH5 [27]. It is important to note that some endolysins have a synergistic effect with antibiotics, which might help to boost their efficacy against chronic infections. Some examples include endolysins P128, CF-301 and ClyS, which turned out to be inhibitory of planktonic and biofilm MRSA cells when combined with oxacillin, probably due to the increase in cell wall permeability mediated by the endolysin [56,57,58].

In summary, this work highlights the potential of phage-derived antimicrobial strategies for the treatment of MRSA infections, and decolonization of asymptomatic carriers. Phages and phage lytic proteins could, therefore, be used to substitute or enhance the action of antibiotics and become powerful allies in the race to limit the spread of antibiotic resistance.

## Figures and Tables

**Figure 1 antibiotics-09-00447-f001:**
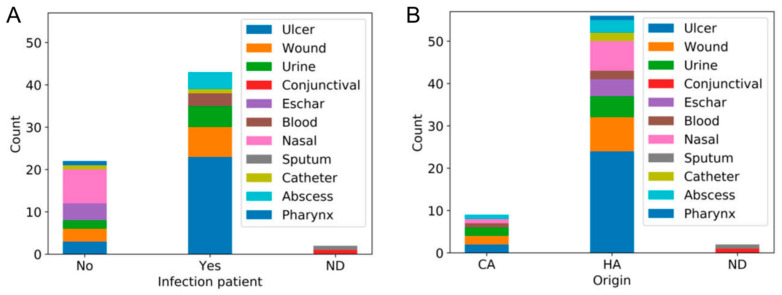
Stacked bar charts representing the number of strains isolated from different sample types that came from patients with active infections or asymptomatic carriers (**A**), and that could be classified as community associated (CA) or hospital associated (HA) (**B**). ND, not determined.

**Figure 2 antibiotics-09-00447-f002:**
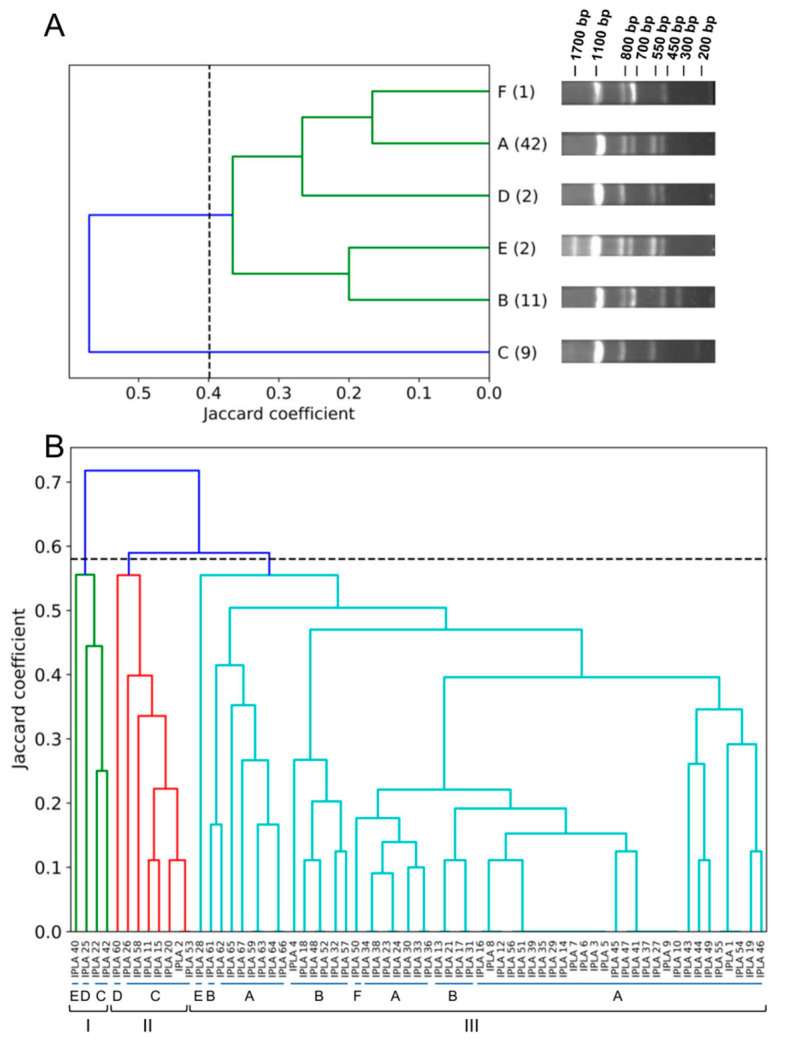
Dendrograms representing clustering analysis of clinical MRSA strains by UPGMA using the Jaccard similarity coefficient. (**A**) Clustering based on RAPD-PCR. The blue line represents strains that were not included in any cluster (isolates with RAPD (Random amplification of polymorphic DNA) profile (C). Separate clusters are shown in different colors. The cut-off value for clustering was 0.399. A, B, C, D, E and F represent the six distinct RAPD profiles. The numbers in brackets indicate how many strains have each profile. (**B**) Clustering based on RAPD profile plus the possession or not of certain virulence and resistance genes. Separate clusters appear in different colors. The cut-off value for clustering was 0.58.

**Figure 3 antibiotics-09-00447-f003:**
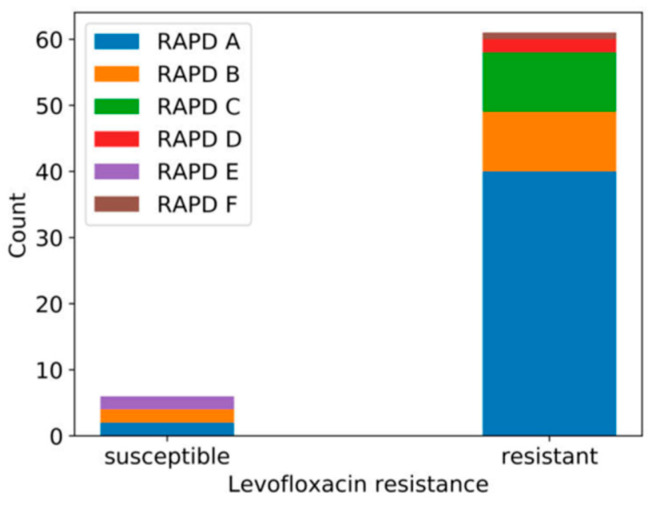
Stacked bar charts representing the number of strains with susceptibility or resistance to levofloxacin that have the different RAPD profiles.

**Figure 4 antibiotics-09-00447-f004:**
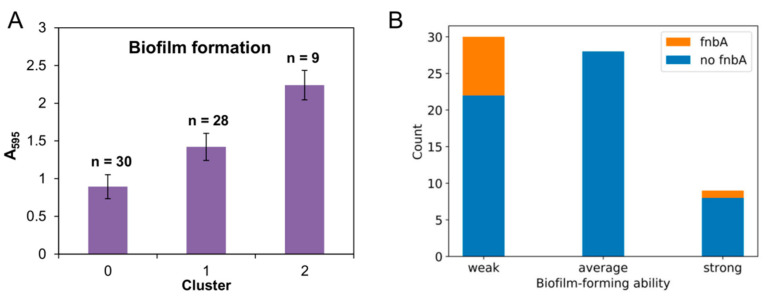
Biofilm formation of clinical MRSA isolates. (**A**) Clustering of MRSA strains based on their biofilm-forming abilities into weak (cluster 0), average (cluster 1) and strong (cluster 2) biofilm formers. The graphs represent the means ± standard deviation of the values corresponding to each group, indicating the number of strains that belong to each cluster. (**B**) Stacked bar chart representing the number of strains for each cluster that carry gene *fnbA* (orange) and those that do not (blue).

**Figure 5 antibiotics-09-00447-f005:**
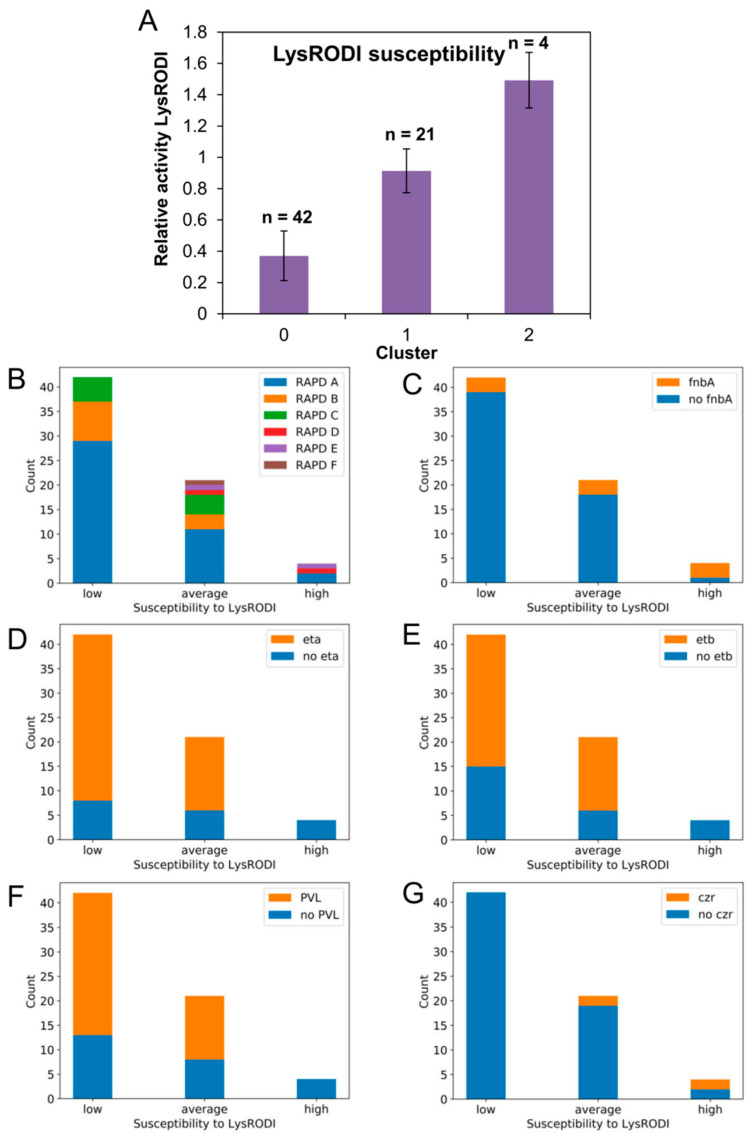
Susceptibility of the clinical MRSA isolates to endolysin LysRODI. (**A**) Clustering of MRSA isolates based on their susceptibility to LysRODI into strains with low (cluster 0), average (cluster 1) and high (cluster 2) susceptibility compared to reference strain Sa9. The graph represents the means ± standard deviations for the values of each group, while also indicating the number of strains that belongs to each cluster. (**B**–**G**) Stacked bar charts representing the number of strains belonging to each cluster that have a different RAPD pattern or carry genes *fnbA*, *eta*, *etb*, PVL and *czr*.

**Table 1 antibiotics-09-00447-t001:** Genotypic and phenotypic characteristics of clinical MRSA isolates.

Strain	Sample	Infection	Origin	RAPD Profile	Susceptibility phiIPLA-RODI	Relative Specific Activity LysRODI *	Biofilm Formation *	Virulence/Resistance Genes *
1	Ulcer exudate	+	HA	A	S	0.869 ± 0.060	1.208 ± 0.736	ebpS, fib, cap5, seg, sei, PVL/Lfx^R^
2	Ulcer exudate	+	HA	C	S	0.778 ± 0.019	1.069 ± 0.611	cna, cap5, sec, seg, sei, eta, etb, PVL/Lfx^R^
3	Ulcer exudate	+	HA	A	S	0.256 ± 0.020	0.627 ± 0.469	ebpS, fib, cap5, seg, sei, eta, etb, PVL/Lfx^R^
4	Ulcer exudate	+	CA	B	S	0.506 ± 0.173	1.084 ± 0.739	fnbB, fib, cap5, see, seg, sei, eta, etb, PVL
5	Urine	+	HA	A	S	0.895 ± 0.053	1.458 ± 0.773	ebpS, fib, cap5, seg, sei, eta, etb, PVL/Lfx^R^
6	Ulcer exudate	+	HA	A	S	0.592 ± 0.058	0.995 ± 0.344	ebpS, fib, cap5, seg, sei, eta, etb, PVL/Lfx^R^
7	Ulcer exudate	+	HA	A	S	0.439 ± 0.047	0.872 ± 0.559	ebpS, fib, cap5, seg, sei, eta, etb, PVL/Lfx^R^
8	Ulcer exudate	+	HA	A	S	0.330 ± 0.040	1.334 ± 0.895	fib, cap5, seg, sei, eta, etb, PVL/Lfx^R^
9	Ulcer exudate	-	HA	A	S	0.345 ± 0.025	1.765 ± 0.618	ebpS, fib, cap5, seg, sei, eta, etb/CLI^R^
10	Ulcer exudate	+	HA	A	S	0.409 ± 0.017	1.506 ± 0.343	ebpS, fib, cap5, seg, sei, eta, etb/Lfx^R^, CLI^R^
11	Conjunctival exudate	ND	ND	C	R	1.136 ± 0.097	0.895 ± 0.552	cna, ebpS, cap5, seg, sei, eta, etb/Lfx^R^
12	Ulcer exudate	+	HA	A	S	0.101 ± 0.030	1.196 ± 0.896	fib, cap5, seg, sei, eta, etb, PVL/Lfx^R^
13	Wound exudate	-	HA	B	S	0.281 ± 0.127	1.353 ± 0.885	ebpS, fib, cap5, seg, sei, eta, etb, PVL/Lfx^R^
14	Ulcer exudate	+	HA	A	S	0.564 ± 0.043	2.236 ± 0.720	ebpS, fib, cap5, seg, sei, eta, etb, PVL/Lfx^R^, CLI^R^
15	Ulcer exudate	+	HA	C	S	0.458 ± 0.044	1.861 ± 0.348	cna, ebpS, cap5, seg, sei, eta, etb, PVL/Lfx^R^*
16	Ulcer exudate	+	HA	A	R	0.380 ± 0.030	0.954 ± 0.389	fib, cap5, seg, sei, eta, etb, PVL/Lfx^R^
17	Ulcer exudate	+	HA	B	S	0.910 ± 0.144	1.375 ± 0.694	ebpS, fib, cap5, seg, sei, eta, etb/Lfx^R^
18	Ulcer exudate	-	HA	B	S	0.619 ± 0.076	0.804 ± 0.496	ebpS, fnbB, fib, cap5, seg, sei, eta, PVL/Lfx^R^
19	Urine	-	HA	A	R	0.681 ± 0.076	2.300 ± 0.324	ebpS, fib, cap5, sed, see, seg, sei/Lfx^R^
20	Eschar exudate	-	HA	C	S	0.569 ± 0.121	1.630 ± 0.796	cna, cap5, seg, sei, eta, etb, PVL/Lfx^R^, CLI^R^
21	Blood culture	+	CA	B	R	0.771 ± 0.099	1.781 ± 0.441	ebpS, fib, cap5, seg, sei, eta, etb, PVL/Lfx^R^
22	Blood culture	+	HA	C	S	0.427 ± 0.109	0.626 ± 0.474	cna, fnbA, cap5, seg, sei, PVL/Lfx^R^
23	Urine	-	CA	A	S	1.068 ± 0.081	0.828 ± 0.291	ebpS, fib, cap5, sed, see, seg, sei, eta, etb, PVL/Lfx^R^
24	Nasal exudate	-	HA	A	R	0.800 ± 0.041	2.214 ± 0.519	ebpS, fib, cap5, sed, see, seg, sei, eta, etb, PVL/Lfx^R^
25	Nasal exudate	-	HA	D	S	1.364 ± 0.153	0.786 ± 0.421	cna, fnbA, fnbB, cap5, seg, sei, czr/Lfx^R^
26	Eschar exudate	-	HA	C	R	0.575 ± 0.062	1.143 ± 0.401	cap5, sed, see, seg, sei, eta, etb, PVL/Lfx^R^
27	Nasal exudate	-	CA	A	R	0.614 ± 0.034	0.739 ± 0.347	ebpS, fib, cap5, seg, sei, eta, etb/Lfx^R^
28	Nasal exudate	-	HA	E	S	1.755 ± 0.169	0.892 ± 0.317	ebpS, fnbA, fib, cap5, sec, seg, sei
29	Sputum	ND	ND	A	S	0.857 ± 0.093	2.287 ± 0.322	ebpS, fib, cap5, seg, sei, eta, etb, PVL/Lfx^R^, CLI^R^
30	Eschar exudate	-	HA	A	S	0.566 ± 0.100	1.384 ± 0.275	ebpS, fib, cap5, sed, see, seg, sei, eta, etb/Lfx^R^, CLI^R^
31	Urine	+	HA	B	S	1.076 ± 0.083	1.160 ± 0.456	ebpS, fib, cap5, seg, sei, eta, etb/Lfx^R^
32	Nasal exudate	-	HA	B	S	0.316 ± 0.054	0.979 ± 0.309	fnbA, fnbB, fib, cap5, seg, sei, PVL/Lfx^R^
33	Wound exudate	+	HA	A	S	1.115 ± 0.311	0.919 ± 0.146	ebpS, fib, cap5, sed, seg, sei, eta, etb/Lfx^R^
34	Ulcer exudate	+	HA	A	R	0.383 ± 0.045	0.693 ± 0.646	ebpS, fib, cap5, sed, seg, sei, eta, etb, PVL/Lfx^R^
35	Nasal exudate	-	HA	A	R	0.456 ± 0.121	1.131 ± 0.360	ebpS, fib, cap5, seg, sei, eta, etb, PVL/Lfx^R^
36	Eschar exudate	-	HA	A	R	0.884 ± 0.048	0.555 ± 0.325	ebpS, fib, cap5, sed, seg, sei, eta, etb/Lfx^R^
37	Wound exudate	-	HA	A	S	0.333 ± 0.035	1.288 ± 0.615	ebpS, fib, cap5, seg, sei, eta, etb/Lfx^R^
38	Pericatheter	+	HA	A	S	0.946 ± 0.008	1.326 ± 0.440	ebpS, fib, cap5, sed, seg, sei, eta, etb, PVL/Lfx^R^
39	Ulcer exudate	+	HA	A	S	0.576 ± 0.061	0.992 ± 0.408	ebpS, fib, cap5, seg, sei, eta, etb, PVL
40	Abscess	+	HA	E	S	1.100 ± 0.105	0.985 ± 0.338	bbp, cna, ebpS, fnbA, cap8, seg, sei, PVL
41	Nasal exudate	-	HA	A	S	0.121 ± 0.019	0.648 ± 0.235	ebpS, fib, cap5, seg, sei, eta, etb/Lfx^R^
42	Ulcer exudate	+	HA	C	S	0.913 ± 0.221	0.893 ± 0.447	cna, fnbA, cap5, cap8, seg, sei/Lfx^R^
43	Abscess	+	HA	A	S	0.363 ± 0.039	0.882 ± 0.484	ebpS, fnbA, fib, cap5, cap8, seg, sei/Lfx^R^
44	Pharyngeal exudate	-	HA	A	S	0.964 ± 0.068	0.865 ± 0.430	ebpS, fnbA, fib, cap5, sed, seg, sei, czr/Lfx^R^
45	Nasal exudate	-	HA	A	S	0.108 ± 0.015	1.571 ± 0.465	ebpS, fib, cap5, seg, sei, eta/Lfx^R^
46	Blood	+	HA	A	S	1.433 ± 0.091	1.043 ± 0.840	ebpS, fib, cap5, sed, seg, sei/LfxR, CLI^R^
47	Urine	+	CA	A	R	0.366 ± 0.047	2.214 ± 0.788	ebpS, fib, cap5, seg, sei, eta, etb/Lfx^R^
48	Abscess	+	HA	B	S	0.102 ± 0.023	1.350 ± 0.353	fnbB, fib, cap5, seg, sei, eta, PVL/Lfx^R^
49	Telescoping catheter	-	HA	A	R	1.418 ± 0.080	2.361 ± 0.863	ebpS, fnbA, fib, cap5, seg, sei, czr/Lfx^R^, CLI^R^
50	Ulcer exudate	+	HA	F	S	0.654 ± 0.126	1.301 ± 0.339	ebpS, fib, cap5, sed, seg, sei, eta, etb, PVL/Lfx^R^
51	Ulcer exudate	-	HA	A	R	0.353 ± 0.052	1.683 ± 0.482	ebpS, fib, cap5, seg, sei, eta, etb, PVL/Lfx^R^
52	Abscess	+	CA	B	S	0.528 ± 0.136	1.404 ± 0.562	fnbB, fib, cap5, seg, sei, eta, PVL/Lfx^R^
53	Ulcer exudate	+	HA	C	S	0.213 ± 0.035	1.000 ± 0.186	cna, cap5, sec, seg, sei, eta, etb, PVL/Lfx^R^
54	Ulcer exudate	+	HA	A	S	0.353 ± 0.035	1.398 ± 0.566	ebpS, fib, cap5, seg, sei, PVL/Lfx^R^, TMP/SMX^R^
55	Ulcer exudate	+	CA	A	R	1.007 ± 0.083	1.204 ± 0.608	ebpS, fib, cap5, seg, sei, PVL/Lfx^R^
56	Ulcer exudate	+	CA	A	S	0.072 ± 0.021	2.583 ± 0.572	ebpS, fib, cap5, seg, sei, eta, etb, PVL/Lfx^R^
57	Wound exudate	+	CA	B	R	0.218 ± 0.031	0.905 ± 0.416	fnbB, fib, cap5, seg, sei, PVL
58	Wound exudate	+	HA	C	S	0.769 ± 0.055	0.879 ± 0.398	cna, cap5, sec, eta, etb, PVL/Lfx^R^
59	Ulcer exudate	+	HA	A	S	0.253 ± 0.031	1.561 ± 0.417	ebpS, fib, cap5, PVL/Lfx^R^
60	Wound exudate	+	HA	D	S	1.001 ± 0.089	1.463 ± 0.147	cna, fnbB, cap5, eta, etb, PVL, czr/Lfx^R^, CLI^R^
61	Wound exudate	+	HA	B	R	0.163 ± 0.030	1.718 ± 0.622	ebpS, fib, cap5, eta, etb/Lfx^R^
62	Ulcer exudate	+	HA	A	R	0.584 ± 0.141	1.223 ± 0.720	ebpS, fib, cap5, eta, etb/Lfx^R^
63	Wound exudate	+	HA	A	R	0.169 ± 0.071	1.307 ± 0.655	ebpS, fib, cap5, PVL/Lfx^R^
64	Urine	+	HA	A	S	0.335 ± 0.013	1.133 ± 0.625	ebpS, fib, cap5, eta, PVL/Lfx^R^, CLI^R^
65	Wound exudate	-	HA	A	R	0.325 ± 0.005	1.280 ± 0.456	ebpS, fib, cap5, sed, eta/Lfx^R^, CLI^R^
66	Urine	+	HA	A	S	0.444 ± 0.358	2.102 ± 0.704	ebpS, fib, cap5, eta, PVL/Lfx^R^
67	Ulcer exudate	+	HA	A	R	0.397 ± 0.024	1.575 ± 0.444	ebpS, fib, cap5/Lfx^R^

* Relative specific activity shows the means and standard deviations of specific activity values of the different strains compared to reference strain Sa9. Biofilm formation shows the means and standard deviations of A595 values from three independent experiments of crystal violet staining. Genes eno, clfA, clfB, icaA, icaD, tst and mecA were present in all the strains. HA: hospital associated origin. CA: community associated origin. Lfx^R^: levofloxacin resistance. CLI^R^: clindamycin resistance. TMP/SMX^R^: trimethopim/sulfamethoxazole (cotrimoxazol) resistance. ND: not determined.

**Table 2 antibiotics-09-00447-t002:** Conditions of the PCR reactions used for the identification of virulence genes.

Virulence factor	Gene (pb)	Primers	Reference
Bone sialoprotein binding protein	*bbp* (575)	BBP-1, BBP-2	[31]
Collagen binding protein	*cna* (423)	CNA-1, CNA-2	
Laminin binding protein	*eno* (302)	ENO-1, ENO-2	
Elastin binding protein	*ebpS* (186)	EBP-1, EBP-1	
Fibronectin binding protein	*fnbA* (643)	FNBA-1, FNBA-2	
	*fnbB* (524)	FNBB-1, FNBB-2	
Fibrinogen binding protein	*fib* (404)	FIB-1, FIB-2	
Clumping factor	*clfA* (292)	CLFA-1, CLFA-2	
	*clfB* (205)	CLFB-1, CLFB-2	
Capsule	*cap5* (361)	Cap5 k1, Cap5 k2	[32]
	*cap8* (173)	Cap8 k1, Cap8 k2	
Polysaccharide matrix	*icaA* (1315)	icaA-R, icaA-R	[33]
	*icaD* (381)	icaD-F, icaD-R	
Proteinaceous matrix	*bap* (971)	BAP-sasp-6m, BAP-sasp-7c	
Enterotoxins	*sea* (127)	SEA3, SEA4	[34]
	*seb* (477)	SEB1, SEB4	
	*sec* (271)	SEC3, SEC4	
	*sed* (319)	SED3, SED4	
	*see* (178)	SEE3, SEE2	
	*seg* (287)	SEG1, SEG2	[35]
	*seh* (213)	SEH1, SEH2	
	*sei* (454)	SEI1, SEI2	
Toxic shock syndrome toxin	*tst* (445)	TST3, TST6	[34]
Exfoliative toxins	*eta* (119)	ETA3, ETA4	
	*etb* (262)	ETB3, ETB4	
Panton-Valentine leukocidin	*PVL* (505)	PVL505-F, PVL505-R	[36]
Cadmium and zinc resistance	*czr* (655)	czrC-F, czrC-R	
Methicillin resistance	*mecA* (264)	mecA264-F, mecA264-R	
Quaternary ammonium compounds resistance	*qacA/B* (192)	QacA/B-F, QacA/B-R	

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
