# Peer review of "Characterization of Clinical MRSA Isolates from Northern Spain and Assessment of Their Susceptibility to Phage-Derived Antimicrobials"

_antibiotics, 2020, doi:10.3390/antibiotics9080447_

Round 1
Reviewer 1 Report
The present study consists of two main topics: i) characterization of MRSA isolates; ii) their susceptibility to two phages and one endolysin. Regarding the latter, the topic is interesting and the study is correctly conducted and well written.
On the contrary, the description of strains characterization is a little messy, so that it is difficult for the reader to accept the final conclusions stated at lines 372-373.
Five of the six RAPD profiles were considered strictly related and grouped in a unique cluster (Fig 2A). Nevertheless, the following comprehensive analysis of typing plus virulence and resistance genes (fig 2B) upsets previous results: isolates belonging to profile C are distributed in all the three clusters (although only one strain falls in the light blue one), seem very similar to the four isolates showing the D and E profiles and distant from those belonging to the B type. This is probably due to the fact that virulence and resistance genes are often located on mobile genetic elements (MGEs), abundant in S. aureus (correctly cited at lines 46-47), and this result simply confirms the frequent exchange of MGEs between strains circulating in the same environment. However, the distribution of the RAPD profiles in figure 2B should result immediately evident to the reader, for example using different colours for strain belonging to different profiles. Moreover, the authors should explain what criteria they used to choose the cut off values (a reference?), as this is pivotal for final conclusion but, at the moment, their choice seems quite arbitrarily.
Major remarks
- Lines 96-118: the paragraph extensively describes the state of the art of some endolysins not related with the present study. It is partially off topic and should be deleted. On the contrary a brief description of some determinants they investigated in the study could be appropriate.
- Legend of figure 2: “The blue line represents strains that were not included in any cluster”: which are these strains? Probably those with RAPD profile C in figure 2A, but it is not clear. The cut off chosen to define “a cluster” should be clearly indicated in the figure (for example with a line) and the reason of the choice should be explained.
- Line 351: “with A being the most abundant” is largely expected, considering that type A includes 63% of the isolates. More relevant could be that sporadic isolates showed only average or high susceptibility to LysRODI, while isolates belonging to the most represented types are distributed between low (mainly) and average susceptibility.
- Overall, a lot of comparisons have been carried out, but not between isolates collected from infection and those from simply colonization: this is not negligible, in my opinion.
Minor remark
- line 179: cut off values used for biofilm formation ability should be defined
- line 241 and 246: Fig 2 instead of Fig 1
- line 253: the number in brackets instead of the figures in brackets
Author Response
Reviewer 1
The present study consists of two main topics: i) characterization of MRSA isolates; ii) their susceptibility to two phages and one endolysin. Regarding the latter, the topic is interesting and the study is correctly conducted and well written.
On the contrary, the description of strains characterization is a little messy, so that it is difficult for the reader to accept the final conclusions stated at lines 372-373.
Five of the six RAPD profiles were considered strictly related and grouped in a unique cluster (Fig 2A). Nevertheless, the following comprehensive analysis of typing plus virulence and resistance genes (fig 2B) upsets previous results: isolates belonging to profile C are distributed in all the three clusters (although only one strain falls in the light blue one), seem very similar to the four isolates showing the D and E profiles and distant from those belonging to the B type. This is probably due to the fact that virulence and resistance genes are often located on mobile genetic elements (MGEs), abundant in S. aureus (correctly cited at lines 46-47), and this result simply confirms the frequent exchange of MGEs between strains circulating in the same environment. However, the distribution of the RAPD profiles in figure 2B should result immediately evident to the reader, for example using different colours for strain belonging to different profiles. Moreover, the authors should explain what criteria they used to choose the cut off values (a reference?), as this is pivotal for final conclusion but, at the moment, their choice seems quite arbitrarily.
Hierarchical clustering is very useful to assess how similar different samples (in this case isolates) are to each other, but unfortunately does not determine the number of clusters. There are different criteria to select a cutoff for dendrograms, although as far as we know none of them is considered optimal. Indeed, in many cases the best way to approach cutoff selection is by looking at the shape of the dendrogram itself (now indicated in the methods section, lines 152-153).
In our case, we first considered the cut-off values given by the clustering algorithm in scipy, which are based on the same formula utilized in MATLAB. The results of this “default” cutoff selection were 0.399 and 0.503 for these two dendrograms:
We left the first dendrogram as it was, giving one large group and one orphan cluster (C strains). However, in the second dendrogram, we considered that the “default” cut-off value left three strains in orphan clusters that did not give any indication of which clusters they were closer to. For this reason, we reckoned that a cut-off value of 0.58 would be more appropriate. With these considerations in mind, we obtained the following dendrogram:
As the reviewer suggested we now indicate the cut-off points in Fig. 2 and in the text. We also indicate the RAPD profile of the different strains in the second dendrogram (Fig. 2B), which we think makes it easier to interpret. It can now be observed that all strains with a C profile are included in the green and red clusters, together with a few isolates from the low frequency profiles. In turn, the cyan cluster comprises all strains with A, B and F profiles, as well as one strain with E profile. Therefore, it appears that the cyan cluster comprises most strains from the large cluster obtained in the first dendrogram, reinforcing the idea that they are genetically similar. In turn, the C strains are now subdivided in two different clusters together with some “rare” isolates. The RAPD band patterns of these “rare” isolates were fairly similar to those in the large cluster, but a closer look to the genes they possess indicates that most of them are actually more similar to the C strains. Taking this into consideration, we do not think that the second dendrogram contradicts the first one, but rather that it complements it, giving further information into the genetic relatedness of the strains.
Major remarks
- Lines 96-118: the paragraph extensively describes the state of the art of some endolysins not related with the present study. It is partially off topic and should be deleted. On the contrary a brief description of some determinants they investigated in the study could be appropriate.
We appreciate the Reviewer’s suggestion. The text was modified accordingly (lines 96-100 and 111-115).
- Legend of figure 2: “The blue line represents strains that were not included in any cluster”: which are these strains? Probably those with RAPD profile C in figure 2A, but it is not clear. The cut off chosen to define “a cluster” should be clearly indicated in the figure (for example with a line) and the reason of the choice should be explained.
We now indicate that the strains not included in any cluster belong to profile C.
Lines marking the cut-off points have been included in both dendrograms.
- Line 351: “with A being the most abundant” is largely expected, considering that type A includes 63% of the isolates. More relevant could be that sporadic isolates showed only average or high susceptibility to LysRODI, while isolates belonging to the most represented types are distributed between low (mainly) and average susceptibility.
We deleted “with A being the most abundant” and included the information suggested by the reviewer. Lines 344-347.
- Overall, a lot of comparisons have been carried out, but not between isolates collected from infection and those from simply colonization: this is not negligible, in my opinion.
We completely agree with the reviewer that it is interesting to assess whether the isolate being isolated from an infected patient or from a carrier has an impact on the other characteristics examined. For this reason, the potential correlation of the variable “infection” and other variables was analyzed. However, we did not find any correlation between isolates collected from an infection or from a carrier and most variables. The only exception to this is the “sample type” as is mentioned in the text and shown in Fig. 1A.
Minor remark
- line 179: cut off values used for biofilm formation ability should be defined
Cut-off values used for biofilm clustering are now defined in the results section.
- line 241 and 246: Fig 2 instead of Fig 1
Done
- line 253: the number in brackets instead of the figures in brackets
Done
Reviewer 2 Report
In this paper, authors well described the significant effect of endolysin (LysRODI) and phiIPLA-RODI phage in counteracting Methicillin-resistant Staphylococcus aureus from both infected patients and asymptomatic carriers in a Spanish hospital. The results achieved are important and have a novelty in the point of the continuous growing awareness of the efficacy of new antimicrobial agents and in the development of efficient targeted therapies against antibiotic-resistant bacteria.
However, there are several issues that should be addressed before publication:
- line 25: “please replace “infection patients” with “infected patients”.
- line 42: please provide some reference supporting the first sentence.
- line 82: : “please replace “selective” with “selectively”.
- lines 99-117: too much information on the topic in this part (particularly commercial), I believe that it should be deleted
- line 230: p > 0.05. when something is not significant then it shouldn't be reported any p value.
- line 231: p > 0.05. when something is not significant then it shouldn't be reported any p value.
- line 282: p > 0.05. when something is not significant then it shouldn't be reported any p value.
- line 308: p > 0.05. when something is not significant then it shouldn't be reported any p value.
- line 319: p > 0.05. when something is not significant then it shouldn't be reported any p value.
- line 344: p > 0.05. when something is not significant then it shouldn't be reported any p value
Author Response
Reviewer 2
In this paper, authors well described the significant effect of endolysin (LysRODI) and phiIPLA-RODI phage in counteracting Methicillin-resistant Staphylococcus aureus from both infected patients and asymptomatic carriers in a Spanish hospital. The results achieved are important and have a novelty in the point of the continuous growing awareness of the efficacy of new antimicrobial agents and in the development of efficient targeted therapies against antibiotic-resistant bacteria.
However, there are several issues that should be addressed before publication:
- line 25: “please replace “infection patients” with “infected patients”.
Done
- line 42: please provide some reference supporting the first sentence.
Done
- line 82: : “please replace “selective” with “selectively”.
Done
- lines 99-117: too much information on the topic in this part (particularly commercial), I believe that it should be deleted
Done
- line 230: p > 0.05. when something is not significant then it shouldn't be reported any p value.
Done
- line 231: p > 0.05. when something is not significant then it shouldn't be reported any p value.
Done
- line 282: p > 0.05. when something is not significant then it shouldn't be reported any p value.
Done
- line 308: p > 0.05. when something is not significant then it shouldn't be reported any p value.
Done
- line 319: p > 0.05. when something is not significant then it shouldn't be reported any p value.
Done
- line 344: p > 0.05. when something is not significant then it shouldn't be reported any p value
Done
Round 2
Reviewer 1 Report
The manuscript has been significantly improved after revision.
I keep on thinking that the decision to increase the cut off value to 0.58 is not really correct, quite arbitrary for a proper epidemiologic analysis. Indeed, the use of the cut off obtained by scipy provided a better description of the relatedness among the isolates tested, identifying three of them as sporadic as, in my opinion, they should be considered.
However, it is true that this does not affect significantly the conclusion of the most important topic, that is the antibacterial activity of phages and endolysin, so, for the aim of this study, it could be considered acceptable.
Minor checks
Many references to figures must be corrected:
- Paragraph 3.2: Fig 2 instead of Fig 1
- Paragraph 3.3: Fig 4 instead of Fig 2
- Paragraph 3.5: Fig 5 instead of Fig 3
Author Response
The manuscript has been significantly improved after revision.
I keep on thinking that the decision to increase the cut off value to 0.58 is not really correct, quite arbitrary for a proper epidemiologic analysis. Indeed, the use of the cut off obtained by scipy provided a better description of the relatedness among the isolates tested, identifying three of them as sporadic as, in my opinion, they should be considered.
However, it is true that this does not affect significantly the conclusion of the most important topic, that is the antibacterial activity of phages and endolysin, so, for the aim of this study, it could be considered acceptable.
Thank you very much for your comments.
Minor checks
Many references to figures must be corrected:
- Paragraph 3.2: Fig 2 instead of Fig 1
Done, lanes 236, 263
- Paragraph 3.3: Fig 4 instead of Fig 2
Done, lanes 288, 304
- Paragraph 3.5: Fig 5 instead of Fig 3
Done, lanes 324, 344, 348, 349, 354